# System Analysis Based on Lipid-Metabolism-Related Genes Identifies AGT as a Novel Therapy Target for Gastric Cancer with Neoadjuvant Chemotherapy

**DOI:** 10.3390/pharmaceutics15030810

**Published:** 2023-03-02

**Authors:** Le Zhu, Ming Ma, Lumin Zhang, Shun Wang, Yu Guo, Xinxin Ling, Hanchao Lin, Nannan Lai, Shengli Lin, Ling Du, Qiongzhu Dong

**Affiliations:** 1Key Laboratory of Whole-Period Monitoring and Precise Intervention of Digestive Cancer, Shanghai Municipal Health Commission (SMHC), Minhang Hospital, Fudan University, Shanghai 201199, China; 2Gastroenterology Department of Minhang Hospital, Fudan University, Shanghai 201199, China; 3Endoscopy Center and Endoscopy Research Institute, Zhongshan Hospital, Fudan University & Shanghai Collaborative Innovation Center of Endoscopy, Shanghai 200032, China

**Keywords:** GC, lipid metabolism, AGT, chemotherapy, prognosis, responsiveness

## Abstract

Gastric cancer (GC) is one of the most common causes of cancer-related deaths worldwide, and chemotherapy is still a standard strategy for treating patients with advanced GC. Lipid metabolism has been reported to play an important role in the carcinogenesis and development of GC. However, the potential values of lipid-metabolism-related genes (LMRGs) concerning prognostic value and the prediction of chemotherapy responsiveness in GC remains unclear. A total of 714 stomach adenocarcinoma patients were enrolled from the Cancer Genome Atlas (TCGA) and Gene Expression Omnibus (GEO) database. Using univariate Cox and LASSO regression analyses, we developed a risk signature based on LMRGs that can distinguish high-GC-risk patients from low-risk patients with significant differences in overall survival. We further validated this signature prognostic value using the GEO database. The R package “pRRophetic” was applied to calculate the sensitivity of each sample from high- and low-risk groups to chemotherapy drugs. The expression of two LMRGs, AGT and ENPP7, can predict the prognosis and response to chemotherapy in GC. Furthermore, AGT significantly promoted GC growth and migration, and the downregulation of AGT enhanced the chemotherapy response of GC both in vitro and in vivo. Mechanistically, AGT induced significant levels of epithelial–mesenchymal transition (EMT) through the PI3K/AKT pathway. The PI3K/AKT pathway agonist 740 Y-P can restore the EMT of GC cells impaired by AGT knockdown and treatment with 5-fluorouracil. Our findings suggest that AGT plays a key role in the development of GC, and targeting AGT may help to improve the chemotherapy response of GC patients.

## 1. Introduction

Gastric cancer (GC), a heterogeneous disease characterized by epidemiologic and histopathologic differences, is the third most common cause of cancer-related death globally [1,2,3]. Despite the decreasing incidence of GC, the prognosis of GC is still unfavorable owing to the frequent presence of stage IV metastatic disease at primary presentation [4]. At present, molecular classification has huge influences on the medical management of GC, which benefits from the development of next-generation sequencing and other genomic technologies [1,5,6]. However, chemotherapy, especially neoadjuvant chemotherapy, still plays a vital role in the GC treatment strategy when it comes to refractory GC patients [7,8,9]. In this way, predicting the responsiveness of chemotherapy in GC appears to be particularly important. 

Lipids constitute the basic structure of biological membranes and function as signaling molecules and an energy source [10,11,12]. A dysregulated lipid metabolism represents an important metabolic alteration in the tumor. Fatty acids, cholesterol, and phospholipids are the three common lipids that act as energy producers, signaling molecules, and source material for the biogenesis of cell membranes [13]. In GC, lipid metabolism has also been shown to be associated with tumor formation and development. For example, the expression of miR-422a can help to drive a metabolic shift from aerobic glycolysis to oxidative phosphorylation by decreasing pyruvate dehydrogenase kinase 2 (PDK2) and thereby modulate de novo lipogenesis [14]. Moreover, tumor-derived PGE2 also promotes tumor progression by suppressing proliferation and increasing the apoptosis of NK cells [15]. Alterations in the lipid metabolisms of tumor cells have gradually become of concern due to the fact that lipid metabolism is closely associated with tumor development [16,17,18]. In addition, it has been reported that lipid metabolism is associated with drug resistance when it comes to treating tumor cells [11,19]. 

Angiotensinogen (AGT), the precursor substrate of the renin–angiotensin–aldosterone system (RAAS) pathway, has been considered to play a vital role in tumorigenesis [20,21]. AGT with variants was first regarded to be associated with the risk of developing an astrocytoma [22]. In a transgenic mouse model of hepatocellular carcinoma, the overexpression of human AGT decreased angiogenesis and delayed tumor progression [23]. However, AGT, which is attributed to tumor proliferation and migration, as well as invasion, has been demonstrated to be a diagnostic and prognostic biomarker in colorectal carcinoma [24]. In GC, the use of AGT as a prognostic molecule has been issued in previous studies [25]. However, the function of AGT in chemotherapy and the underlying mechanism of AGT are poorly understood in the context of GC. 

Hence, in this study, we conducted a large-scale analysis to identify the clinical risk factors of lipid-metabolism-related genes (LMRGs) for the prediction of the prognosis and responsiveness of chemotherapy in GC, which provides an efficient classification tool for patient stratification in GC. Following this, a series of in vitro and in vivo experiments were conducted to evaluate the potential role of AGT in tumor proliferation and migration and the efficacy of chemotherapy. 

## 2. Materials and Methods

### 2.1. Patients and Cohorts

All public gene expression data and corresponding clinical characteristics were extracted from the Gene Expression Omnibus (GEO) (https://www.ncbi.nlm.nih.gov/geo/) and The Cancer Genome Atlas (TCGA) (https://portal.gdc.cancer.gov/) database. Cohorts of 359 samples and 355 samples corresponding with clinical information from TCGA and GSE84433, respectively, were enrolled in this study. 

### 2.2. Extraction of Genes in Lipid Metabolism

Lipid-metabolism-related sets were downloaded from the GSEA/MSigDB (http://www.gsea-msigdb.org/gsea/). The details of each set of genes are documented in Appendix A. After removing overlapping genes, 831 LMRGs were obtained.

### 2.3. Functional Analyses

For biological process and pathway enrichment analyses, the Kyoto Encyclopedia of Genes and Genomes (KEGG) and Gene Ontology (GO) analyses were performed using the R “clusterProfiler” package. 

### 2.4. Chemotherapeutic Response Prediction

Based on the Genomics of Drug Sensitivity in Cancer (GDSC) database, we performed chemotherapeutic response prediction for each STAD sample. Four commonly used drugs were selected, namely, Cisplatin, Doxorubicin, 5-Fluorouracil and Mitomycin C. The R package “pRRophetic” was utilized for calculation, and the half-maximal inhibitory concentration (IC50) was estimated [26].

### 2.5. Establishment and Validation of Risk Model

The “limma” package [27] was used to compare the differentially expressed genes (DEGs) between chemotherapy response and non-response groups. The threshold value was set to |log2 (foldchange)| > 1, *p* < 0.05, for further analysis. The same threshold value was also used to filter DEGs among the generated differentially expressed genes by comparing AGT-high and AGT-low groups with chemotherapy. Least absolute shrinkage and selection operator (LASSO) analysis was conducted to downsize the prognostic genes previously filtrated using the “glmnet” R package. The merged LMRGs containing AGT and ENPP7, which were derived from chemotherapy-related DEGs and 29 genes sorted via LASSO analysis, were enrolled in the risk model. The risk score of each patient in the training and verification cohorts was calculated as risk score = 0.0207 × expression value of AGT + 0.1248 × expression value of ENPP7, and patients were divided into low-risk and high-risk groups. Kaplan–Meier analysis and ROC analysis were used to test the suitability and stability of the model. 

### 2.6. RNA Interference and Plasmid Transfection

While the cells reached 80% confluence, transfection was conducted according to the lipofectamine 3000 (Invitrogen, Waltham, MA, USA) instructions. Lentiviral short hairpin RNA (shRNA) targeting AGT (CCACCTTCATACCTGCTCCAA for sh1, and CCAGGAGTTCTGGGTGGACAA for sh2), REG3A (CCTGGTGAAGAGCATTGGTAA for sh1, and CCGAGCCCAATGGAGAAGGTT for sh2), and Sh-control (Scr) obtained from Sigma-Aldrich were loaded into the PLKO plasmid. The AGT and REG3A CDS sequences were loaded into the PCDH plasmid for AGT and REG3A overexpression in GC cells.

### 2.7. Cell Viability Assay

CCK8 test kits were employed to test cell viability following the manufacturer’s instructions. After being seeded in 96-well plates, cancer cells were incubated with DMEM containing 10% CCK8 for 2 h, and the absorbance was measured at 450 nm.

### 2.8. Western Blot

The protein concentration was determined using a BCA kit after cells were lysed in RIPA buffer with a phosphatase inhibitor cocktail. Then, the proteins were boiled with loading buffer at 100 °C for 20 min. Proteins were loaded and separated via electrophoresis on SDS-polyacrylamide gel electrophoresis (SDS-PAGE) and transferred to a PVDF membrane. The membrane was incubated with primary antibodies against AGT (ProteinTech Group, Chicago, IL, USA), E-cadherin (CST, Danvers, MA, USA), N-cadherin (CST, Danvers, MA, USA), Vimentin (CST, Danvers, MA, USA), PI3K (CST, Danvers, MA, USA), Phospho-PI3K (Affinity, Nanjing, Jiangsu, China), AKT (CST, Danvers, MA, USA), Phospho-AKT (CST, Danvers, MA, USA), REG3A (Abcam, Cambridge, MA, USA), and β-actin (CST, Danvers, MA, USA) at 4 °C overnight. Then, the membrane was further incubated with secondary antibody for one hour at room temperature, and enhanced chemiluminescence (ECL) reagent was used to visualize the protein bands using a Gel Doc EZ Imager. β-actin was used as an internal reference.

### 2.9. Colony Formation Assay

Transfected HGC-27 and BGC-823 cells (1000 cells/well) were seeded into 6-well plates and cultured for 2 weeks. Finally, colonies were counted after being fixed with 4% paraformaldehyde and stained with 0.1% crystal violet.

### 2.10. Cell Migration

Cells at the logarithmic growth phase were harvested and resuspended in serum-free medium to be seeded in Transwell chambers for detecting their migration ability. After 24 h or 48 h, the Transwell chambers were collected. Cells were fixed with 4% paraformaldehyde and stained with crystal violet. The numbers of migrating cells were counted under a light microscope.

### 2.11. ELISA

The supernatants of BGC-823 cell lines in various groups were harvested after 48 h and subjected to a commercial human Ang-I ELISA kit (Multi Sciences, Hangzhou, China) according to the manufacturer’s protocols. The OD450 values of reaction were measured using a multifunction microplate reader (Synergy/HTX, BioTek). The concentrations of Ang-I were calculated by referring to the corresponding standard curve.

### 2.12. Establishment of GC Subcutaneous Xenograft Models in Nude Mice

Human gastric cancer cells, including Sh-control HGC-27 and Sh-AGT HGC-27 (2 × 106/100 μL in 50% Matrigel and PBS per mice), were injected into the right flank of nude mice (5 mice/group) to establish the subcutaneous implantation models. For 5-Fu treatment, experiments involved a dose of 20 mg/kg 5-Fu or 100 µL PBS (for control), administrated intraperitoneally with one dose every 3 days. The tumor growth was monitored once per week until the mice were sacrificed. Tumor volume was calculated using the following formula: a × b2/2 (a and b represent the largest and smallest tumor diameters, respectively). All of the tumors were collected, and the tumors’ weights were measured.

### 2.13. Statistical Analyses

Mann–Whitney U tests was employed to compare two groups, whereas the Kruskal–Wallis test and one-way analysis were used to compare more than two groups [28]. Correlation coefficients were computed via Spearman and distance correlation analyses. The cutoff values of each dataset were based on the association between patient overall survival and risk score using the “survminer” package. R software (version 4.0.3) or SPSS software (version 25.0) were employed for all statistical analyses.

## 3. Results

### 3.1. Development of Risk Model with Prognosis and Response to Chemotherapy Based on LMRGs in Stomach Adenocarcinoma 

The workflow of this study is depicted in Figure 1. We collected a total of 714 GC patients from the TCGA and GEO databases, including 359 stomach adenocarcinoma (STAD) patients from the TCGA database and 355 patients from the GEO database. The detailed clinicopathological information of all of the included patients is presented in Table 1. To identify prognosis-related genes from LMRG signatures in GC, we first performed univariate Cox regression analysis using 359 STAD patients in the TCGA database and identified that 95 LMRGs were significantly associated with GC prognosis (Appendix A). 

Next, a risk signature model was constructed to evaluate the prognostic prediction of LMRGs in gastric cancer. LASSO analysis was conducted to screen potential genes for establishing a risk model, and 29 genes were filtered using an optimal lambda value (Appendix A). The established risk model classified the gastric cancer patients into low-risk and high-risk groups. The patients in the low-risk group had a better OS compared to the patients in the high-risk group (Figure 2A). Regarding the diagnosis of the risk model, ROC analysis showed that the constructed risk model exhibited precise predictive capacity over a period of 5 years, and the area under the curve (AUC) of the ROC curve for 1, 3, and 5 years was 0.57, 0.61, and 0.67, respectively (Appendix A). 

Chemotherapy is still a standard strategy used to treat patients with advanced GC [29]. Therefore, we further compared the differentially expressed genes (DEGs) between the chemotherapy response group and the chemotherapy non-response group based on the TCGA database. In total, 197 genes were found to be downregulated, and 162 genes were upregulated (Appendix A). Based on the merged genes generated from the DEGs of chemotherapy and risk-model-related genes through LASSO analysis, AGT and ENPP7 were overlapped among these signatures. The expression boxplot displayed the discrepancy between the low-risk and high-risk groups, and the high-risk group showed a high level expression of AGT and ENPP7 (Figure 2B,C). Meanwhile, the high-risk group had a higher mortality rate and shorter survival time compared with the low-risk group (Figure 2D). Consistently, the cohorts of low AGT and ENPP7 expression displayed good prognosis compared to the high-expression group (Appendix A).

To elucidate the association between lipid metabolism and chemotherapy response in GC, we further explored the prediction of risk signature on chemotherapy response in the TCGA dataset. The Genomics of Drug Sensitivity in Cancer (GDSC) database was used to predict the chemotherapy response of patients in GC. Four conventionally used drugs for GC, including Cisplatin, Doxorubicin, 5-Fluorouracil, and Mitomycin C, were used for the analysis. The results showed that the estimated IC50 for each of the four agents was significantly lower in the low-risk group than that in the high-risk group (Figure 3A–D, *p* < 0.001), indicating that GC patients with higher risk scores tended to be more resistant to chemotherapy than those with lower risk scores. To evaluate whether GC patients with a low-risk signature might benefit more from chemotherapy compared with GC patients with a high-risk signature, we investigated the association between risk signature and OS among GC patients who either did or did not receive chemotherapy. The results confirm that patients treated with chemotherapy have a higher rate of OS in the group with low risk signatures (Figure 3E).

### 3.2. Prognostic Value and Validation of the Risk Signature

To evaluate the clinical significance of the risk model, we further explored the association between the risk score and clinical features in the GEO database. Patients were divided into low- and high-risk groups using the best cutoff value through a function named “surv_cutpoint” based on the survival status, survival time, and obtained risk score. The survival analysis revealed that patients in the high-risk group showed an inferior survival compared with those in the low-risk group (Figure 4A). The expression levels of AGT and ENPP7 were consistently higher in the high-risk groups compared with the low-risk group (Figure 4B,C). Meanwhile, the high-risk group had a higher mortality rate and shorter survival time compared with the low-risk group (Figure 4D). Since clinicopathological characteristics may also possess a prognostic value, a nomogram integrating the risk signature and clinical features was constructed to predict the prognosis of GC in the TCGA dataset. The constructed nomogram indicated that risk score and chemotherapy situation were endowed specific scores based on their contributions to the prognosis of GC (Figure 5A). Then, we validated the nomogram in the TCGA and GEO cohorts. The observed overall survival matched relatively well with the actual survival at 5 years in the TCGA cohort (Figure 5B), and a similar result was also observed in the GEO cohort (Figure 5C). 

### 3.3. AGT Promotes GC Proliferation and Migration

Previous studies have reported that AGT was associated with poor prognosis in GC [25,30]. To investigate the functions of AGT in GC, we knocked down the expression of AGT in HGC-27 and BGC-823 cells via LV-Sh-AGT. AGT knockdown resulted in the significant suppression of the cell proliferation (Figure 6A,B), clone formation (Figure 6C,D), and migration ability of HGC-27 and BGC-823 cells (Figure 6E,F). In contrast, exogenous AGT expression in HCG-27 and BGC-823 cells significantly enhanced their cell proliferation, clone formation, and migration capacity compared with the negative control (Appendix A). To further determine the effects of AGT on in vivo tumor growth, HGC-27 cells stably transfected with Sh-AGT or scrambled shRNA were injected subcutaneously into nude mice. The average tumor volume of HGC-27 cells stably transfected with Sh-AGT was significantly smaller than tumors in the control group (Figure 6G).

### 3.4. AGT Activated EMT via PI3K/AKT Pathway in GC Cells

The epithelial–mesenchymal transition (EMT) plays a very important role in the process of cancer development and tumor metastasis [31]. Chemoresistance, an opposite property to chemotherapy responsiveness, has been demonstrated to be associated with the EMT signaling pathway [32]. To further determine whether AGT is involved in the EMT of GC, we evaluated EMT markers in HGC-27 and BGC-823 cells with AGT knockdown. A reduced expression of N-cadherin and vimentin concomitant with a significant increase in E-cadherin were found after AGT knockdown (Figure 7A). On the contrary, the overexpression of AGT induced a decrease in the E-cadherin level and significant increases in the expression levels of N-cadherin and vimentin (Figure 7B). In addition, the morphological changes in cell characteristics of the EMT phenotype are shown in Appendix A. We overexpressed AGT in the HGC-27 cell line and found that the upregulation of AGT resulted in morphological changes in GC cells from the typical cobblestone-like appearance of epithelial cells to the fibroblastic morphology.

To determine how AGT induces EMT, GO and KEGG enrichment analyses were performed on DEGs derived from the comparison between the AGT-high and AGT-low groups (Figure 7C,D, Appendix A). Interestingly, the PI3K/AKT signaling pathway was remarkably enriched (Figure 7D). The PI3K/AKT pathway has been demonstrated to be involved in the EMT of several types of human cancer [33,34,35,36]. To explore whether PI3K/AKT signaling was involved in AGT-mediated EMT, we examined the effect of AGT on the activation of PI3K/AKT. The phosphorylation of PI3K and AKT was notably enhanced by AGT overexpression, whereas the knockdown of AGT significantly decreased the phosphorylation of PI3K and AKT (Figure 7E,F, Appendix A). Consistently, the activation of the PI3K/AKT pathway by its specific activator (740 Y-P, 20μM) markedly attenuated the expression of E-cadherin and increased the expression of N-cadherin and Vimentin in GC cells with AGT knockdown (Figure 7G,H). These results indicated that the PI3K/AKT pathway is critical in the EMT induced by AGT. 

### 3.5. REG3A Mediated the Activation of the PI3K/AKT Pathway by AGT in GC Cells 

Since AGT is the precursor substrate of the RAAS pathway, we investigated whether RAAS was involved in malignant phenotypes induced by AGT. We first performed an ELISA assay to detect the concentration of Angiotensin I (Ang-I), which was derived from the AGT, in the supernatant of GC cell lines and found that AGT had no effect on the expression of angiotensin-I (Ang-I) in GC cells with an overexpression or knockdown of AGT (Appendix A). Then, we analyzed the correlation between AGT and Angiotensin II Receptors (AGTRs) in the TCGA database using the GEPIA web tool. No significant correlation was found between AGT and AGTRs (Appendix A). We used the RT-qPCR assay to examine the expression of AGTR1 and AGTR2 in the BGC-823 cell line and found no significant difference between the Vector and OE-AGT group (Appendix A). Furthermore, we treated GC cells with Valsartan, an oral antagonist of angiotensin receptor [37], to evaluate the malignant phenotypes. Surprisingly, Valsartan treatment had no effect on the proliferation induced by AGT (Appendix A). Based on the results indicating that the function of AGT is independent on RAAS in GC, we made DEGs between the high- and low-AGT expression level groups. The top five DEGs are presented in a volcano plot in Appendix A. In order to explore the most significant gene that correlated with AGT in GC, we also performed an RT-qPCR assay in HGC-27 and BGC-823 cells and found that REG3A was the hub gene in the downstream signaling of AGT (Appendix A). An increased expression of REG3A was found after AGT knockdown, and the overexpression of AGT induced a decrease in the REG3A expression level (Figure 8A,B). Furthermore, the phosphorylation of PI3K and AKT was notably decreased by REG3A overexpression, whereas the knockdown of REG3A significantly enhanced the phosphorylation of PI3K and AKT (Figure 8C,D). Next, the Sh-AGT/Sh-REG3A or OE-AGT/OE-REG3A were co-transduced in HGC-27 and BGC-823 cells. The results showed that REG3A is involved in the process of the AGT/PI3K/AKT axis as a mediator (Figure 8E,F). 

### 3.6. Targeting AGT Promoted the Efficacy of Chemotherapy of GC through Inhibiting EMT via the PI3K/AKT Signaling Pathway

Next, to gain more insights into the association between chemotherapy efficacy and AGT, we detected the proliferation and tumor growth of GC cells with Sh-AGT and/or treated with 5-fluorouracil. The knockdown of AGT impeded the proliferation of GC cells. In addition, the cell proliferation could also be inhibited when GC cells were treated with 5-fluorouracil, especially in GC cells with Sh-AGT (Figure 9A,B). The migration capacities were also notably decreased in HGC-27 and BGC-823 cells either with Sh-AGT or treatment with 5-fluorouracil. Moreover, we found that the knockdown of AGT in combination with 5-fluorouracil led to a more significant decrease in the migration of GC cells than that with the use of single interventions (Figure 9C,D). 

To further evaluate the effect of AGT on chemotherapy, we established subcutaneous GC models in our study. The results demonstrated that HGC-27-Vector xenografts grew faster than both HGC-27-Sh-AGT xenografts and HGC-27 xenografts when treated with 5-fluorouracil (Figure 6G). In addition, an additive effect was exerted when we treated the HGC-27- Sh-AGT xenografts with 5-fluorouraci (Figure 10A,B). 

To explore the role of the EMT regulated by the PI3K/AKT signaling pathway in chemotherapy, GC cells with knockdown of AGT were treated with the PI3K/AKT activator (740 Y-P, 20 μM) and/or 5-fluorouracil. The migration ability of GC cells eliminated during treatment with 5-fluorouracil was obviously restored with the addition of 740 Y-P (Figure 9E,F). Meanwhile, the Western blot results showed that 740 Y-P could reverse the activation of the EMT signaling pathway with an increased expression of N-cadherin as well as vimentin and a decreased E-cadherin expression level in GC Sh-AGT cells or those treated with 5-fluorouracil (Figure 7G,H). These data indicate that AGT could inhibit chemotherapy efficacy by regulating EMT via activating the PI3K/AKT pathway in GC cells.

## 4. Discussion

Lipids, which mainly consist of fatty acids, cholesterol, and phospholipids, act as energy providers, signaling molecules, and source materials for the biogenesis of cell membranes [38]. The lipid metabolism plays a vital role in cancer progression by not only performing functions between tumor cells and immune-associated cells but also by favoring tumor cell survival and proliferation under harsh conditions [13]. An increasing number of studies has demonstrated that lipid metabolism plays a crucial role in tumor development and that tumor progression is associated with the rewiring of lipid metabolism in tumor cells as well as tumor-surrounded immune cells in the context of tumors [39,40,41]. 

Since neoadjuvant chemotherapy plays an important role in treating advanced and resectable GC patients [42,43], we enrolled four common chemotherapeutic drugs to assess their responsiveness between low-risk and high-risk groups through the R package “pRRophetic”. The half-maximal inhibitory concentration (IC50) of each chemotherapeutic drug in all samples showed that the risk score is positively correlated with the concentration of each drug. Furthermore, we performed a gene expression differential analysis between the chemotherapy response and non-response groups. Next, we linked the DEGs generated from the above comparison and the LMRGs obtained from LASSO regression analysis. Then, we used the two-merged LMRGs containing AGT and ENPP7 to construct the risk model. The superior prognosis of patients in the low-risk group compared with the patients in the high-risk group demonstrated the reliability of this risk model. Moreover, we sorted patients treated with chemotherapeutic drugs for further prognosis analysis based on the two generated LMRGs. Both AGT and ENPP7 play an important role in lipid metabolism. For one thing, AGT, which is the precursor substrate of the RAAS, can be cleaved to angiotensin (Ang) [44]. In transgenic rats, circulating Ang is positively associated with lipid and glucose metabolism [45]. Furthermore, Ang II is produced by adipose tissue, which reflects the role of this hormone in the regulation of fat mass and associated disorders [46]. Furthermore, ENPP7, also named Alkaline sphingomyelinase, has been found to be a key enzyme in hydrolyzing sphingomyelin in the gut. ENPP7 was also reported to be a physiological factor promoting cholesterol absorption by reducing sphingomyelin levels in the intestinal lumen [47]. In colon cancer, intestinal ENPP7 is localized in the enterocytes, which may have antiproliferation effects on tumor cells [48]. 

Notably, AGT has been reported to be associated with poor prognosis and occurrence in GC [25]. Targeting AGT has been widely used to suppress tumor proliferation, migration, and invasion [24]. In addition, the combination of silencing AGT with checkpoint blockage generated an abscopal effect in resistant tumors [49]. We made DEGs between AGT-high and AGT-low groups based on the TCGA database concerning GC with chemotherapy. The GO and KEGG showed that lipid metabolism is associated with the extracellular region and extracellular part, protein digestion and absorption, Staphylococcus aureus infection, the PI3K/AKT signaling pathway, and so on. For instance, the PI3K/AKT pathway, which is involved in the regulation of tumor growth and chemoresistance in GC, was identified as a marker predicting the infiltration of immune cells and the efficacy of immunotherapy [50,51]. The extracellular region includes many components, among which the extracellular matrix was found to modulate the stiffness of tissue through regulating the activation of an oncogenic gene named the Yes-associated protein (YAP) in GC [52]. The lipid metabolism can also increase the sensitivity of T-cell-mediated tumor killing in such a way that increases tumor immunogenicity by elevating antigen presentation [53]. Furthermore, many studies have reported that EMT is one of the crucial processes in promoting tumor metastasis and cancer development [54,55], and EMT has been demonstrated to impact drug resistance in various cancers [54,56]. Our study demonstrated that AGT could promote the expression of EMT-related markers and that the PI3K/AKT agonist 740 Y-P can restore the EMT of GC cells impaired by AGT knockdown. These findings suggest that the PI3K/AKT signaling pathway may be a key factor in AGT modulation of the EMT. AGT is one of the players of the RAAS pathway that regulates blood pressure and fluid balance. Up until now, the RAAS pathway hasn’t been reported to be associated with the tumor malignant phenotype. Continuously, we also did not find evidence that RAAS contributed to the function of AGT in GC. The role of RAAS in cancer may be an interesting area for cancer research in the future. In this study, we also found the mediator named REG3A between the AGT and PI3K/AKT pathways in GC. Interestingly, REG3A was reported to suppress gastric cancer cell invasion and proliferation through the PI3K/AKT signaling pathway [57]. 

Unsurprisingly, the combination of immunotherapy and chemotherapy has been popularized in treating advanced GC patients [3,58]. In a clinical trial, the combination of the chemotherapy drug CapeOx plus anti-PD-1 sintilimab obtained an encouraging pathological complete response (pCR) rate and a good safety profile in a neoadjuvant setting [59]. When comparing the clinical efficacy between sintilimab and a placebo in combination with chemotherapy, sintilimab and chemotherapy significantly improved OS and progression-free survival compared with the placebo and chemotherapy treatment in patients with advanced or metastatic esophageal squamous cell carcinoma [60]. Considering the extensive usage of chemotherapeutic drugs in GC patients, we applied this risk signature in predicting the responsiveness of chemotherapy. Despite deep learning radiomics being developed to predict patient response to neoadjuvant chemotherapy, it is still difficult to guarantee its reliability and accuracy owing to its single-source diagnostic information [61]. Our in vitro and in vivo study indicated that the combination of targeting AGT and chemotherapy can obtain an additive effect.

Notably, previous studies have found that AGT was correlated with GC progression and serves as a prognostic biomarker [25,62]. Our findings solidify these analyses and unravel the function of AGT in the chemotherapy response of GC patients. Interestingly, we found that targeting AGT may enhance the efficacy of the chemotherapy of GC, suggesting that AGT may be a prognostic biomarker for GC patients and could serve as a potential target to increase the efficacy of chemotherapy for GC treatment.

## 5. Conclusions

In conclusion, this study defined a risk signature from the perspective of the lipid metabolism based on comprehensive bioinformatics analysis. Moreover, the risk signature remained significant in GC patients with chemotherapy. The two LMRGs, AGT and ENPP7, may serve as novel biomarkers to predict prognosis and response to chemotherapy in GC. The inhibition of AGT may enhance the chemotherapy response of GC. Our findings indicated that the risk signature, especially AGT, may help to predict the prognosis and contribute to the personalized management of GC patients.

## Figures and Tables

**Figure 1 pharmaceutics-15-00810-f001:**
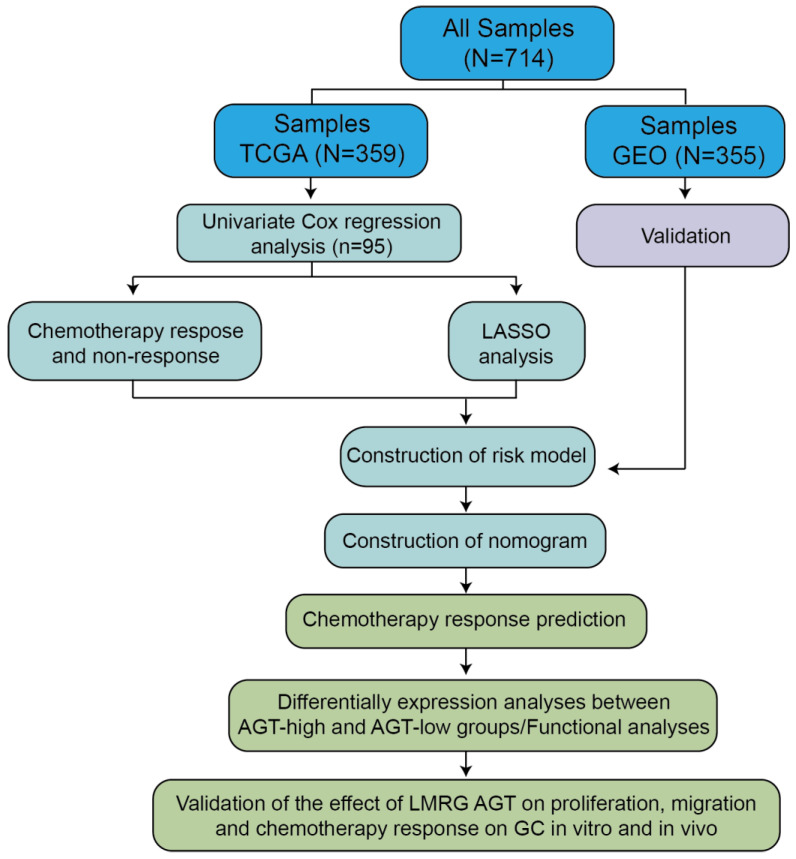
Flowchart of the overall procedures. The processes of data collection and analyses for prognostic studies and chemotherapy response are illustrated.

**Figure 2 pharmaceutics-15-00810-f002:**
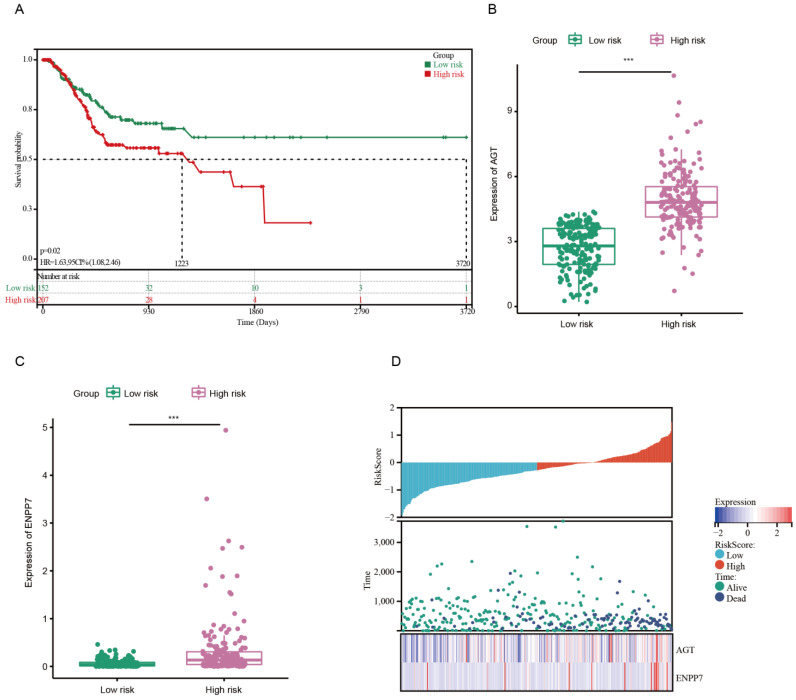
Construction of risk model in the TCGA cohort. (**A**) Kaplan–Meier curves of OS in the TCGA cohort based on risk score. (**B**,**C**) Expression of AGT and ENPP7 in low-risk and high-risk groups. (**D**) Risk score and expression heatmap of AGT and ENPP7 in GC from TCGA. *** *p* < 0.001.

**Figure 3 pharmaceutics-15-00810-f003:**
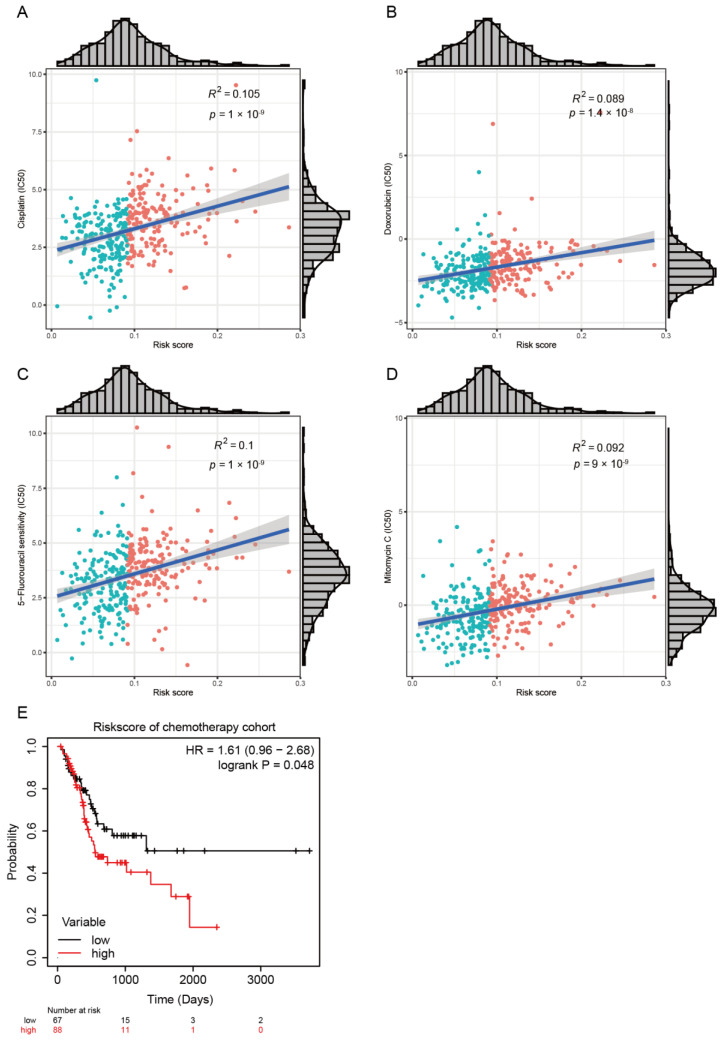
Impact of risk score on chemotherapy response. The estimated half-maximal inhibitory concentration (IC50) of Cisplatin (**A**), Doxorubicin (**B**), 5-Fluorouracil (**C**), and Mitomycin C (**D**) for response between the low-risk and high-risk groups. (**E**) Kaplan–Meier curves of OS in the chemotherapy-treated patients based on risk score.

**Figure 4 pharmaceutics-15-00810-f004:**
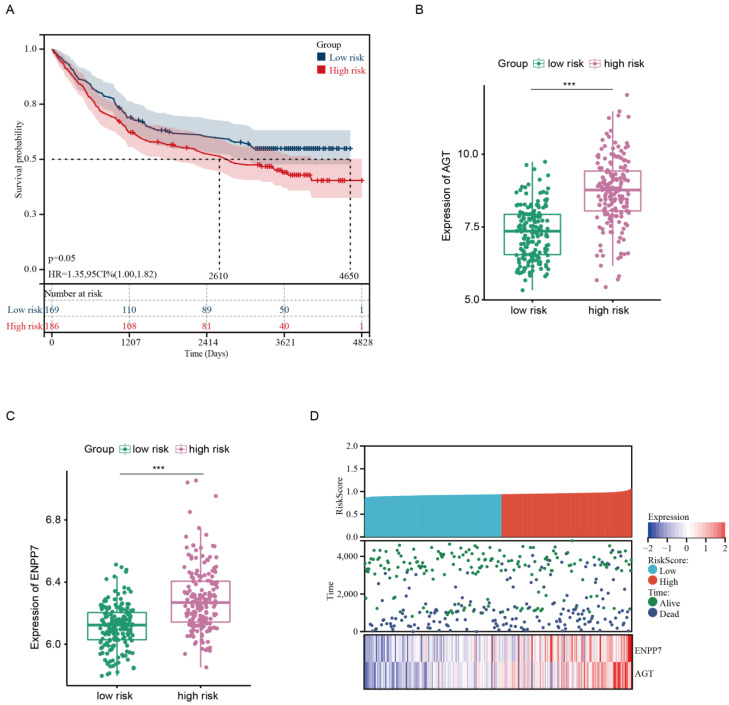
Validation of risk model in GEO cohort. (**A**) Kaplan–Meier curves of OS in the GEO cohort based on risk score. (**B**,**C**) Expression of AGT and ENPP7 in low-risk and high-risk groups. (**D**) Risk score and expression heatmap of AGT and ENPP7 in GC from GEO. *** *p* < 0.001.

**Figure 5 pharmaceutics-15-00810-f005:**
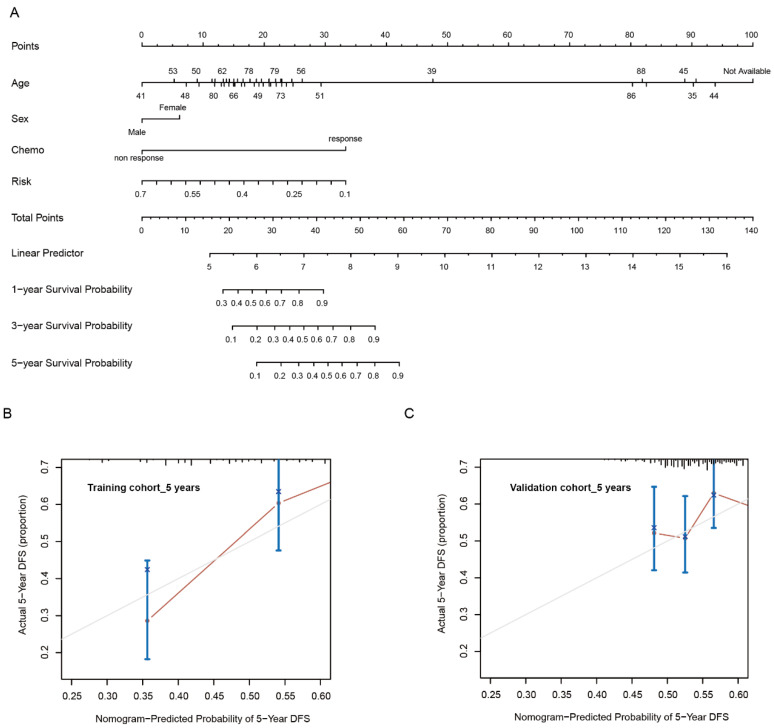
Construction and calibration of nomogram. (**A**) Nomogram integrating risk score and clinical features. (**B**,**C**) Calibration of the nomogram at 5 years in the training cohort and in the verification cohort.

**Figure 6 pharmaceutics-15-00810-f006:**
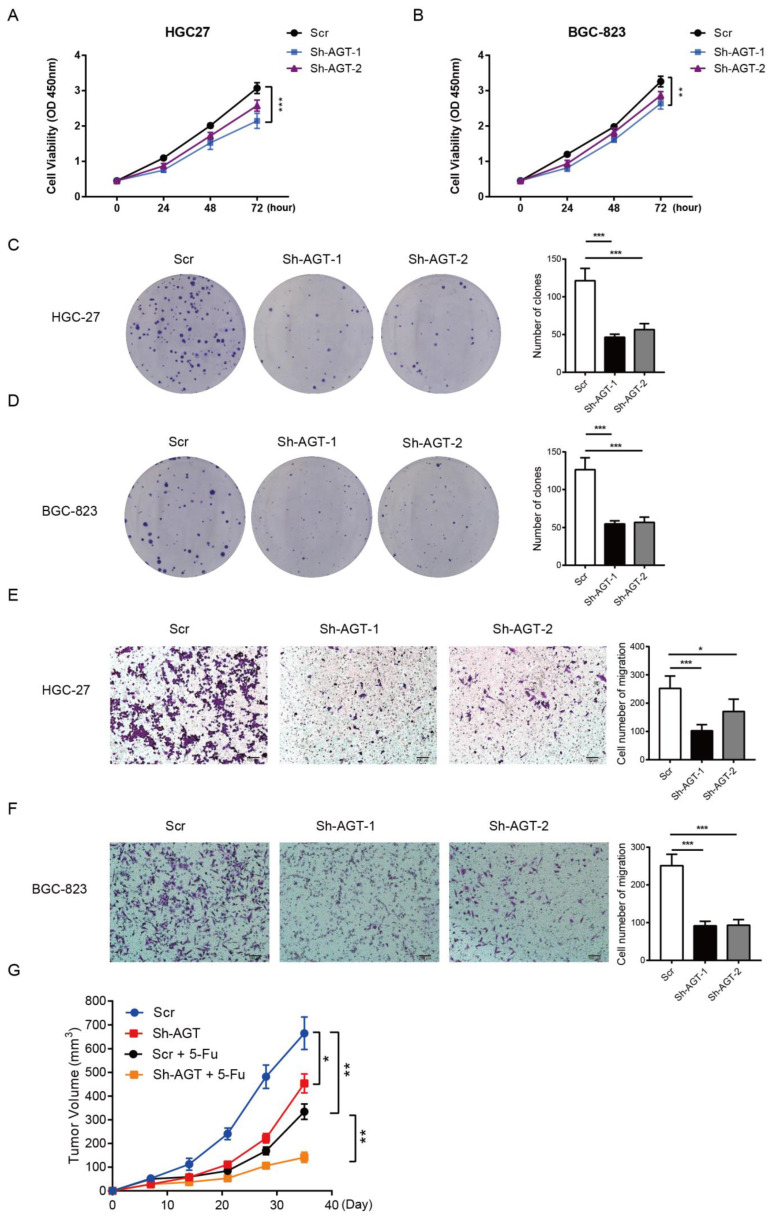
AGT promoted cell proliferation and cell migration in GC cells. (**A**,**B**) Cell viability was determined via CCK8 assay in the HGC-27 and BGC-823 cell lines transfected with Scr or two independent Sh-AGTs. (**C**,**D**) Colony formation assay of HGC-27 and BGC-823 cell lines transfected with Scr or two independent Sh-AGTs. (**E**,**F**) HGC-27 and BGC-823 cells were grown and transiently transfected with Scr or two independent Sh-AGTs and then subjected to Transwell for 24 h. Scale bar = 100 μm. (**G**) The tumor growth of xenograft tumors from HGC-27 with Sh-AGT and 5-Fu. n = 5, mean ± SEM, *** *p* < 0.001, ** *p* < 0.01, * *p* < 0.05. Sh, short hairpin; Scr, Sh-control; and 5-Fu, 5-fluorouracil.

**Figure 7 pharmaceutics-15-00810-f007:**
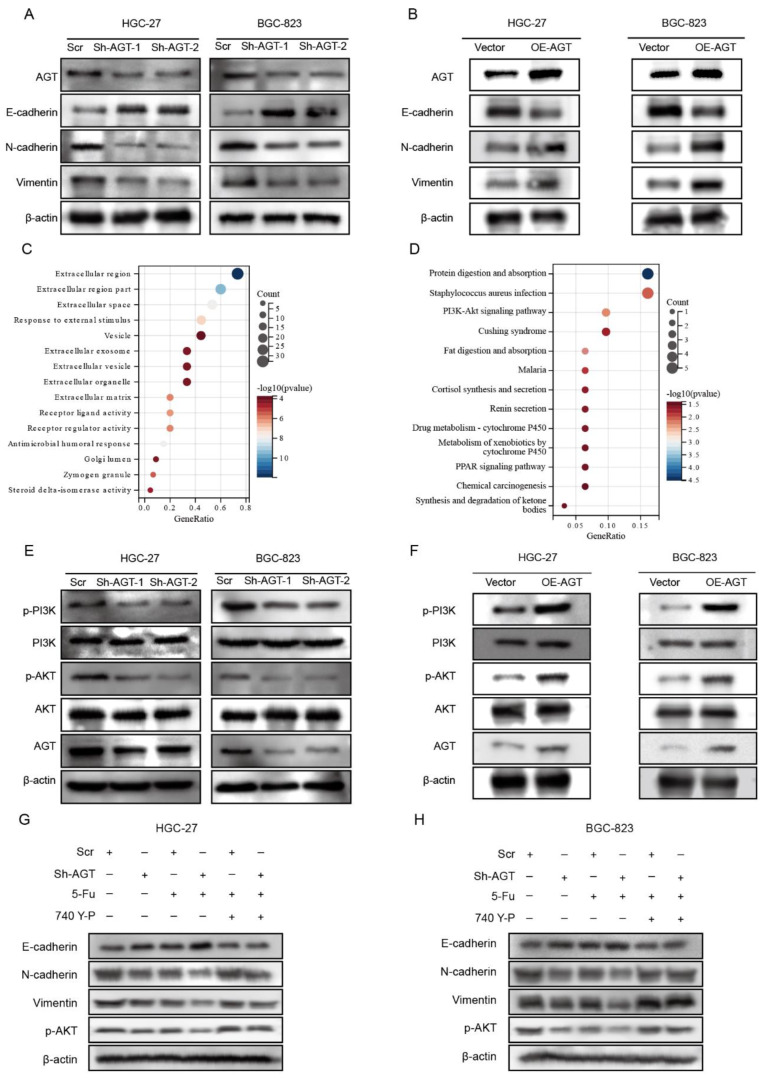
AGT activated the EMT and the PI3K/AKT pathway in GC cells. (**A**) Relative expression levels of E-cadherin, N-cadherin, Vimentin, AGT, and β-actin in the HGC-27 and BGC-823 cells transfected with Scr or Sh-AGT. (**B**) Relative expression levels of E-cadherin, N-cadherin, Vimentin, AGT, and β-actin in the HGC-27 and BGC-823 cells transfected with Vector or OE-AGT plasmid. (**C**,**D**) GO and KEGG analyses for differenced genes between AGT-high and AGT-low groups. (**E**) Relative expression levels of p-PI3K, PI3K, p-AKT, AKT, AGT, and β-actin in the HGC-27 and BGC-823 cells transfected with Scr or Sh-AGT. (**F**) Relative expression levels of p-PI3K, PI3K, p-AKT, AKT, AGT, and β-actin in the HGC-27 and BGC-823 cells transfected with Vector or OE-AGT plasmid. (**G**,**H**) Relative expression levels of E-cadherin, N-cadherin, Vimentin, p-AKT, and β-actin in the indicated cells treated with or without 5-Fu and 740 Y-P.

**Figure 8 pharmaceutics-15-00810-f008:**
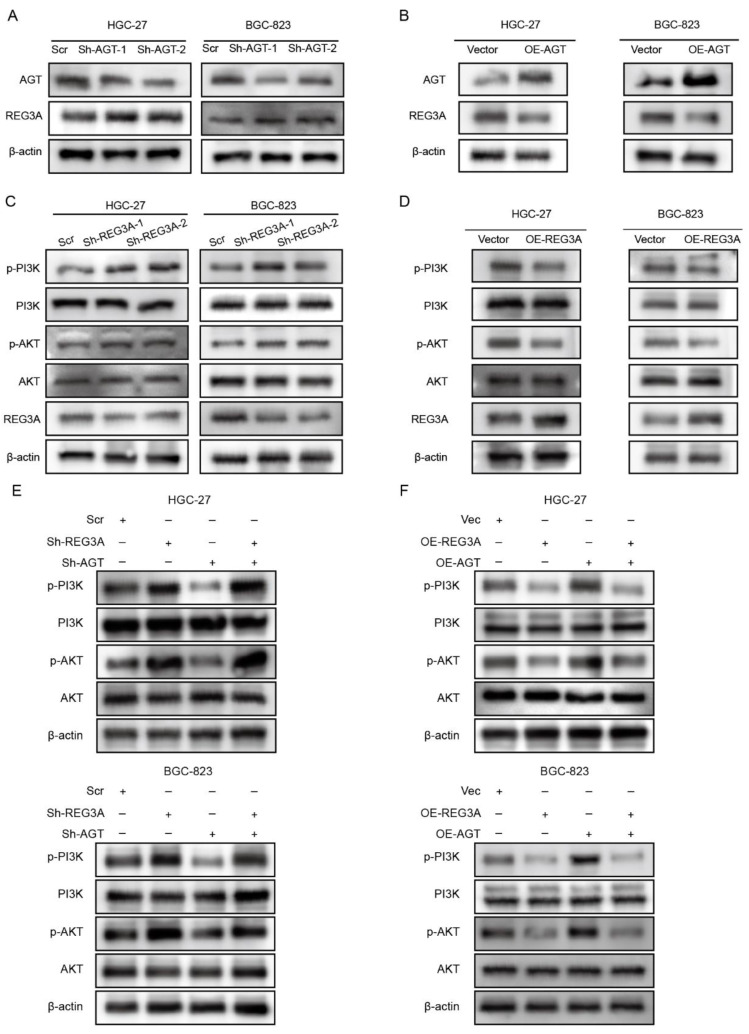
REG3A involved in AGT/PI3K/AKT signaling axis in GC cells. (**A**) Relative expression levels of AGT, REG3A, and β-actin in the HGC-27 and BGC-823 cells transfected with Scr or Sh-AGT. (**B**) Relative expression levels of AGT, REG3A, and β-actin in the HGC-27 and BGC-823 cells transfected with Vector or OE-AGT plasmid. (**C**) Relative expression levels of p-PI3K, PI3K, p-AKT, AKT, REG3A, and β-actin in the HGC-27 and BGC-823 cells transfected with Scr or Sh-REG3A. (**D**) Relative expression levels of p-PI3K, PI3K, p-AKT, AKT, REG3A, and β-actin in the HGC-27 and BGC-823 cells transfected with Vector or OE-REG3A plasmid. (**E**) Relative expression levels of p-PI3K, PI3K, p-AKT, AKT, and β-actin in the indicated cells transfected with or without Sh-REG3A and Sh-AGT. (**F**) Relative expression levels of p-PI3K, PI3K, p-AKT, AKT, and β-actin in the indicated cells transfected with or without OE-REG3A and OE-AGT plasmid.

**Figure 9 pharmaceutics-15-00810-f009:**
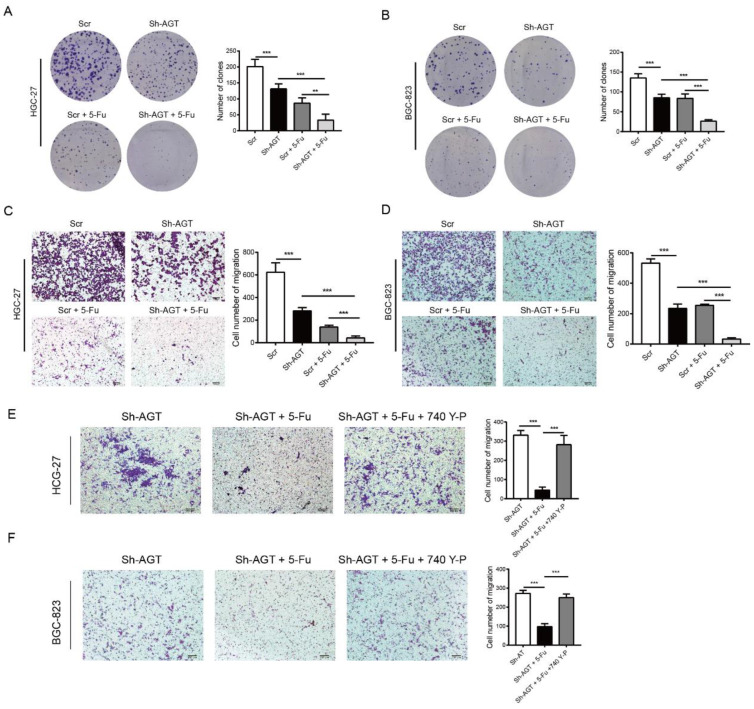
Targeting AGT promoted chemotherapy efficacy through inhibiting EMT in GC cells via the PI3K/AKT signaling pathway. (**A**,**B**) Images and quantified analysis of clone formation assay in HGC-27 and BGC-823 cells with Sh-AGT and 5-Fu. (**C**,**D**) Representative images and quantified analyses of Transwell assay in HGC-27 and BGC-823 cells with Sh-AGT and 5-Fu (2 × 10^4^ cells, 48 h). (**E**,**F**) Representative data from Transwell assay performed in the indicated cells treated with or without 5-Fu or 740 Y-P (HGC-27 and BGC-823, 2 × 10^4^ cells, 48 h). Scale bar = 100 μm. *** *p* < 0.001, ** *p* < 0.01.

**Figure 10 pharmaceutics-15-00810-f010:**
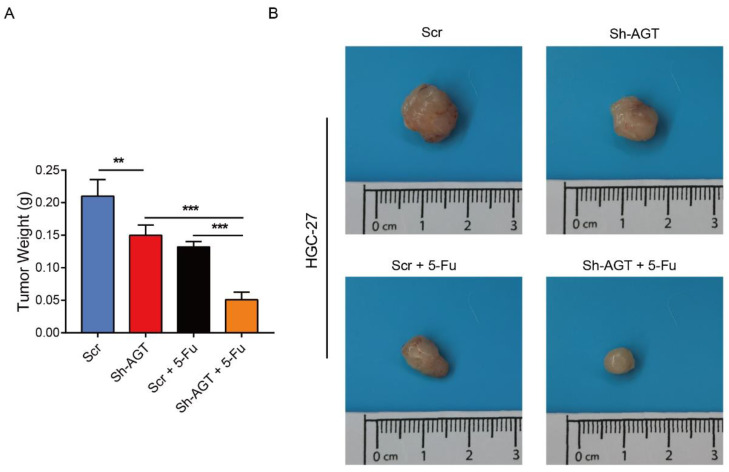
Sh-AGT group showed better chemotherapy efficacy on gastric cancer than Scr group in vivo. (**A**,**B**) The tumor weight (**A**) and final representative images (**B**) of xenograft tumors from HGC-27 with Sh-AGT and 5-Fu. n= 5, mean ± SEM, *** *p* < 0.001, ** *p* < 0.01.

**Table 1 pharmaceutics-15-00810-t001:** Patients and tumor clinicopathological characteristics of 714 gastric cancer patients involved in the study.

Characteristics	All (N = 714)	Detailed Data	
		TCGA (N = 359)	GEO batch (N = 355)
Age			
<65	374 (52.38%)	147 (40.95%)	227 (63.94%)
≥65	340 (47.62%)	212 (59.05%)	128 (36.06%)
Gender			
Male	471 (65.97%)	229 (63.79%)	242 (68.17%)
Female	243 (34.03%)	130 (36.21%)	113 (31.83%)
Status			
Alive	408 (57.14%)	225 (62.67%)	183 (51.55%)
Dead	306 (42.86%)	134 (37.33%)	172 (48.45%)
TNM stage			
T1	19 (2.66%)	8 (2.23%)	11 (3.10%)
T2	128 (17.93%)	93 (25.91%)	35 (9.86%)
T3	220 (30.81%)	155 (43.17%)	65 (18.31%)
T4	339 (47.48)	95 (26.46%)	244 (68.73%)
TX	8 (1.12%)	8 (2.23%)	0 (0.00%)
Chemotherapy			
Response	—	73 (47.10%)	—
Non-response	—	50 (32.25%)	—
Unavailable	—	32 (20.65%)	—

## Data Availability

Please contact the author for data requests.

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
