# Peer review of "System Analysis Based on Lipid-Metabolism-Related Genes Identifies AGT as a Novel Therapy Target for Gastric Cancer with Neoadjuvant Chemotherapy"

_pharmaceutics, 2023, doi:10.3390/pharmaceutics15030810_

Round 1
Reviewer 1 Report
Authors performed in silico analysis of lipid related genes potentially involved in gastric cancer (GC) prognosis. Authors found that angiotensinogen (AGT) and ENPP7 may participate in GC malignancy. Indeed, they found by in vitro and in vivo experiments that AGT plays roles in GC progression. Further, they showed by animal experiments that targeting AGT may be beneficial for adjuvant chemotherapy. This study is potentially important but I have concerns as listed below.
Major issues
1) Introduction: No description about relationship between AGT and GC although many papers seem to be published about this issue. Authors should show what is new in this study? Are there any published data regarding cancer cell-derived AGT and its functions?
2) Authors showed that cancer cell-derived AGT can enhance malignant phenotype of GC cells via PI3K-Akt pathway. However, it is totally unclear how AGT is processed in GC cells and functions to mediate PI3K-Akt signaling. Further experiments need to be done to solve this key issue.
3) EMT data shown in Fig 7A-F are not convincing. Experiments should be repeated (at least total 3 times) and quantitatively evaluated.
4) Page 14, line 8: “whereas knockdown of AGT significantly decreased the phosphorylation of PI3K” Authors should not use “significantly” without statistical evaluation.
5) Fig 7: Morphological changes of cells characteristic to EMT phenotype should be shown.
6) Page 15: “In addition, a synergistic effect was exerted when we treated” I thought that effect of sh-AGT and 5-FU shown in Fig 9A was additive rather than synergistic.
Other issues
1) Introduction: Relationship between GC and lipid metabolism is unclear. Are there any characteristic relationships between them?
2) Page 6: Description of Figure 3 appeared before discussing Figure 2B. Figures need to be rearranged.
3) Page 8: It is helpful if authors briefly explain how AGT and ENPP7 relate to lipid metabolism.
4) Labeling of x-axis in Fig 6A, B, G is missing.
5) Discussion: They did not discuss ENPP7. Why did they focus on AGT rather than ENPP7 in this study?
Reviewer 2 Report
I read with great interest the work of Zhu et al, which identified the lipid metabolism related gene AGT as a key oncogene in the growth and metastasis of gastric cancer. The authors did extensive in silico, in vitro and in vivo analysis and drew their pertinent conclusions based on solid data. That said, few minor issues need to be addressed:
1. In table 1, it is not clear how the authors calculated the numbers and percentages; for example, in the last three rows under the subheading chemotherapy, the same % of responsive, nonresponsive and unavailable are repeated for ALL vs. TCGA, is this correct?
2. In figure 6G, the there is no control for the Scr group itself (un-manipulated cells) to test whether transfection with the Scr itself has any effect on tumor growth.
3. The manuscript must undergo extensive language editing as it is replete with syntax and phrasing errors. For example, the word mechanically in the abstract should read like mechanistically. See also the image below which shows the first paragraph in the introduction
Round 2
Reviewer 1 Report
Authors extensively revised manuscript and I found that it is improved. However, some issues remain to be solved.
1) Page 6, Section 3.4: “whereas the knockdown of AGT significantly decreased the phosphorylation of PI3K and Akt” This statement needs to refer Figure R1. Figure R1 should be presented as a main figure.
2) Page 7, Section 3.5: Authors added new experimental results showing involvement of REG3A in AGT-PI3K/Akt pathway in response to reviewer comment #2. However, following issues are still unclear.
a. Could AGT produced in GC cells be converted to angiotensin? If so, can authors answer molecular mechanisms? Do authors consider that AGT has functions that are independent of renin-angiotensin system?
b. Authors should examine expression of angiotensin receptor and whether angiotensin receptor antagonist could cancel malignant phenotypes shown in this manuscript.
Round 3
Reviewer 1 Report
I felt that manuscript was further improved because authors responded to all concerns raised by this reviewer. They tested involvement of angiotensin and its receptor on GC phenotypes. It was revealed that angiotensin is not involved in AGT-driven phenotypes. I think that this is intriguing because AGT itself has functions to drive GC progression. However, they showed these results only in response letter. I suggest that figures R1 and R2 should be referred in the manuscript and discuss angiotensin-independent mechanisms may contribute to GC phenotypes. Am I wrong?
